# Nanoscale Strontium-Substituted Hydroxyapatite Pastes and Gels for Bone Tissue Regeneration

**DOI:** 10.3390/nano11061611

**Published:** 2021-06-19

**Authors:** Caroline J. Harrison, Paul V. Hatton, Piergiorgio Gentile, Cheryl A. Miller

**Affiliations:** 1School of Clinical Dentistry, The University of Sheffield, 19 Claremont Crescent, Sheffield S10 2TA, UK; c.j.harrison@sheffield.ac.uk (C.J.H.); c.a.miller@sheffield.ac.uk (C.A.M.); 2School of Engineering, Stephenson Building, Newcastle University, Newcastle upon Tyne NE1 7RU, UK; piergiorgio.gentile@newcastle.ac.uk

**Keywords:** nanoscale calcium phosphate, strontium, injectable biomaterial, bone graft substitute

## Abstract

Injectable nanoscale hydroxyapatite (nHA) systems are highly promising biomaterials to address clinical needs in bone tissue regeneration, due to their excellent biocompatibility, bioinspired nature, and ability to be delivered in a minimally invasive manner. Bulk strontium-substituted hydroxyapatite (SrHA) is reported to encourage bone tissue growth by stimulating bone deposition and reducing bone resorption, but there are no detailed reports describing the preparation of a systematic substitution up to 100% at the nanoscale. The aim of this work was therefore to fabricate systematic series (0–100 atomic% Sr) of SrHA pastes and gels using two different rapid-mixing methodological approaches, wet precipitation and sol-gel. The full range of nanoscale SrHA materials were successfully prepared using both methods, with a measured substitution very close to the calculated amounts. As anticipated, the SrHA samples showed increased radiopacity, a beneficial property to aid in vivo or clinical monitoring of the material in situ over time. For indirect methods, the greatest cell viabilities were observed for the 100% substituted SrHA paste and gel, while direct viability results were most likely influenced by material disaggregation in the tissue culture media. It was concluded that nanoscale SrHAs were superior biomaterials for applications in bone surgery, due to increased radiopacity and improved biocompatibility.

## 1. Introduction

Annually, millions of patients worldwide require grafts or alloplastic biomaterials to repair bone defects caused by trauma, cancer surgery, or congenital deformities. When bone healing is compromised, there can be a great detrimental impact on a patient’s quality of life as well as increased treatment costs. The current gold standard material is an autologous bone graft typically taken from the iliac crest of the patient. However, harvesting of the bone has significant disadvantages in terms of donor-site morbidity, pain suffered by the patient, and graft availability. Bone allograft is therefore commonly employed, but this has associated risks, including viral or prion transmission and a less predictable host response due to processing [1]. Therefore, there is a great interest in improved synthetic biomaterials for effective and consistent bone tissue regeneration. The ideal bone augmentation material should be osteoconductive to aid the regrowth of bone tissue, and be available in a suitable form to allow for the surgeon to easily implant it into the defect site. 

Synthetic calcium phosphates, such as hydroxyapatite (HA), have been widely used in medicine and dentistry. Nanoscale hydroxyapatite (nHA) can be considered bioinspired due to its similarity to the mineral found naturally in bone and tooth enamel [2,3]. Calcium phosphates have typically been used in the form of powders, granules, or as coatings on the surface of implants. However, the development of nHA has allowed alternative forms of bone graft substitute to be investigated. One promising technology that has been developed is a paste based upon the combination of nHA with water. Clinical and animal data for the first generation of nanoscale calcium phosphate paste products (including NANOSTIM (Medtronic) and ReproBone^®^ *novo* (Ceramisys Ltd. Sheffield, UK)) are encouraging, suggesting that they are capable of promoting bone tissue regeneration [4,5]. The reasons underpinning their good clinical performance may be due, in part, to the extremely high surface area-to-volume ratio and bioinspired use of nanoscale calcium phosphates. In detail, it has been suggested that bioinspired elements, such as nanoscale dimensions, calcium deficiency and carbonate substitution [6], may also contribute to a more favourable biological response [2,7]. Advantageously, these systems are injectable, thereby facilitating minimally invasive surgeries, and nanoscale calcium phosphates may be chemically modified to enhance their clinical performance. Despite this potential, modified nanoscale calcium phosphates have had little impact on the market, or clinically. There is also a lack of laboratory studies that have considered the substitution of other chemical elements into injectable nanoscale calcium phosphate systems. Strontium-substituted hydroxyapatite (SrHA) is of particular interest, due to its ability to enhance bone regeneration through the stimulation of osteoblasts, inhibition of osteoclasts [8], and encouragement of mesenchymal stromal cell osteogenic differentiation [9]. Furthermore, it has been shown that the presence of strontium-substituted biphasic calcium phosphate decreased pro-inflammatory cytokine production, suggesting that Sr may assist in controlling inflammatory processes [10]. Recent studies regarding injectable SrHA cement [11] and SrHA integrated with phosphoserine-tethered poly(epsilon-lysine) dendrons [12] have suggested that these materials are promising candidates for bone regeneration purposes, but neither of these studies investigated the full range of possible Sr substitution (i.e., up to 100 at.% Sr).

Sr can substitute into the hydroxyapatite crystal structure in the place of Ca, which causes a linear expansion of the lattice constants [6]. The resulting biological effects may be due to the presence of Sr distorting the crystal lattice, which contributes to a greater solubility for increasing Sr concentrations when compared to Ca hydroxyapatite [13]. The majority of published reports have only investigated limited levels of Sr incorporation [8,9,14,15]. Alternatively, the following reports have investigated the material properties of SrHA produced over the whole range of possible substitution levels, but without characterising their biological effects [16,17]. Whilst there have been some reports describing the biological effects of SrHA over a range of Sr compositions, to date these relied on production via a single synthesis route [18,19]. 

Due to the different Sr ranges investigated and the different methods used to prepare the materials, it is impossible to fully appreciate the effect of the synthesis route on the materials produced and their biocompatibility. By investigating only one production method, it can be difficult to conclude whether a Sr concentration is optimal, or if production effects and the final form of the material are also responsible for differences in the biological effects observed. Similarly, there are very limited reports on injectable SrHA systems. For instance, Raucci et al. investigated SrHA gels, but only up to 20 atomic (at.)% Sr incorporation [9]. Other early reports have also focussed on low levels of Sr incorporation into SrHA [8,20,21]. Since the solubility of SrHA increases with higher Sr amounts [13], investigations into higher Sr concentrations, e.g., 50 and 100 at.% SrHA, may more clearly demonstrate how differences in solubility affect not only the properties of the biomaterial system, but also the preparation of these materials. There is clear evidence in the literature that different methods of nanoscale hydroxyapatite production can have a significant effect on the chemical and physical properties [22]. Therefore, there is a need for investigations that report the production and comparison of SrHA manufactured using alternative synthesis routes to produce consistent materials in a form suitable for delivery via a minimally invasive method, such as injection.

A variety of methods are available to produce biomimetic nanoscale hydroxyapatite, including hydrothermal and wet precipitation [22,23,24]. Advantageously, wet precipitation methods do not require specialised equipment to achieve high pressures and temperatures associated with the hydrothermal method. The authors have previously reported a rapid-mixing wet precipitation method [24] for the preparation of nanoscale hydroxyapatite. This method has also been used to prepare silver-doped nanoscale hydroxyapatite pastes with antibacterial properties [25], thereby demonstrating its versatility for the preparation of chemically modified materials. It has not, however, yet been evaluated for the fabrication of novel injectable SrHA pastes or gels. The aim of this study was therefore to fabricate systematic series (0, 2.5, 5, 10, 50, and 100 atomic% Sr) of SrHA pastes and gels using two different methodological approaches, wet precipitation and sol-gel, and fully characterise the different products. On completion of this research, a reliable method for the preparation of consistent SrHAs will have been identified, suitable for adoption by the medical devices or related biopharma industry.

## 2. Materials and Methods

All reagents were obtained from Sigma-Aldrich (Gillingham, UK) unless otherwise stated. 

### 2.1. Strontium-Substituted Nanoscale Hydroxyapatite (SrHA) Paste Preparation Using Rapid-Mixing Wet Precipitation 

A rapid-mixing method to prepare nanoscale calcium hydroxyapatite previously published by the authors [24] was modified to prepare a range of strontium-substituted nanoscale hydroxyapatite (SrHA: 0, 2.5, 5, 10, 50 and 100 at.% Sr). Specifically, the appropriate amount of calcium hydroxide (Ca(OH)_2_, ≥96% purity) was added alongside the appropriate amount of strontium hydroxide octahydrate (Sr(OH)_2_·8H_2_O, 95% purity) to 500 mL deionised water (Table 1). The calcium/strontium hydroxide suspension was then stirred for 1 h at 400 rpm. 

A phosphoric acid solution (H_3_PO_4_, 30 mmol phosphoric acid (85 wt.% in water, 99.99% purity) dissolved in 250 mL deionised water) was poured into the calcium/strontium hydroxide suspension and the suspension was stirred for a further 1 h. The suspensions were left to settle for approximately 20 h. The SrHA suspensions were then washed at least three times with 500 mL deionised water. Additional washes were carried out until the conductivity was below 15 µS·cm^−1^, measured using FG3 FiveGo™ portable conductivity meter (Mettler Toledo™, Leicester, UK). However, washing was discontinued if the conductivity did not decrease by 10 µS·cm^−1^ with 4 subsequent washes. 

In order to produce a powder for materials characterisation the SrHA suspensions were dried at 60 °C in a drying oven and ground in an agate mortar and pestle. Half of the powder sample was sintered in order to investigate the thermal stability of the materials. The ground powder was placed in a furnace (UAF 15/5, Lenton) with a ramp rate of 10 °C·min^−1^ to a temperature of 1000 °C, which was maintained for 2 h. 

In order to prepare a paste for biocompatibility testing the SrHA suspensions were dried at 60 °C in an oven until the paste had a water content of 80 wt.%. This water content was selected due to the formation of a stable paste suspension that had good injectable properties when loaded into a syringe. The pastes were then stored in sealed universal tubes, sterilised using 25 kGy gamma irradiation from a cobalt 60 source (Swann-Morton Ltd., Sheffield, UK) and loaded into 1 mL syringes prior to use.

### 2.2. Strontium-Substituted Nanoscale Hydroxyapatite Gel Preparation Using Rapid-Mixing Sol-Gel Method 

SrHA gels (0, 2.5, 5, 10, 50 and 100 at.% Sr) were prepared using a rapid-mixing sol-gel method. Briefly, the appropriate amount of calcium nitrate tetrahydrate (Ca(NO_3_)_2_·4H_2_O, >99% purity) was dissolved alongside the appropriate amount of strontium nitrate (Sr(NO_3_)_2_, >99% purity) in 500 mL deionised water (Table 2). Ammonium phosphate dibasic ((NH_4_)_2_HPO_4_, ≥98% purity, 3.96 g (30 mmol)) was dissolved in 250 mL deionised water. The pH of the calcium/strontium nitrate solution and the ammonium phosphate solutions were then adjusted to 11 and 12, respectively, using 1 M potassium hydroxide (KOH, >85% purity) solution. This was measured using the pH 211 microprocessor pH meter (Hanna Instruments). The phosphorus solution was poured into the calcium/strontium solution, which was then stirred for 1 h at 400 rpm and the SrHA suspension was left to settle for approximately 20 h. The SrHA suspensions were then washed and dried using the same parameters as described for the wet precipitation method above. Due to excessive thermal decomposition at 1000 °C, 0 and 100 at.% SrHA were sintered at 700 °C for 2 h using the same parameters as described for the wet precipitation method above, in order to investigate their thermal stability.

SrHA gels of 90 wt.% water content were prepared as this method allowed for the formation of a stable gel-like suspension with a higher water content (90 wt.%) than the suspensions formed by the rapid-mixing wet precipitation method. This allowed for the comparison of injectable SrHA materials with different water contents. The gels were stored and sterilised in the same manner as described for the SrHA pastes.

### 2.3. Material Characterisation 

#### 2.3.1. X-ray Diffraction (XRD)

SrHA powder samples were prepared for XRD by mixing with a small amount of PVA glue on an acetate film, and dried using a hot air gun. Qualitative phase analysis was performed on powdered samples using X-ray diffraction (STOE IP, Darmstadt, Germany) with Cu K_α1_ radiation, λ = 0.15406 nm. The diffractometer was operated at 40 kV and 35 mA, with a 2θ range of 20°–60°. The resulting patterns were analysed using ICDD PDF 4+ software, with the following ICDD cards used for phase identification: 9-432—calcium hydroxyapatite, 04-016-2959—2.4 at.% SrHA, 04-016-3585—5 at.% SrHA, 60-0648—10 at.% SrHA, 34-479—50 at.% SrHA, 33-1348—100 at.% SrHA, 04-014-2292—β-tricalcium phosphate (β-TCP), 24-1008—β-tristrontium phosphate (β-TSP). 

#### 2.3.2. Transmission Electron Microscopy (TEM)

Powder samples were placed in ethanol and dispersed using ultrasound for 15 min, before being pipetted onto a 400-mesh copper grid coated with a holey thin carbon film (Agar Scientific, Stansted, UK). Electron microscopy was carried out in the Faculty of Science, Department of MBB, The University of Sheffield. Morphological evaluation of the SrHA particles was carried out using the Tecnai G2 Spirit TEM (FEI, Hillsboro, USA) at an accelerating voltage of 80 kV. In order to obtain aspect ratios for the particles produced, the lengths and widths of 10 particles were measured per sample. 

#### 2.3.3. X-ray Fluorescence (XRF)

SrHA powder (0.8 g) was combined with 8 g of lithium tetraborate (ICPH, Malzéville, France). This mixture was melted into a glass disc in a furnace at 1200 °C. The glass discs were analysed in an XRF spectrometer (PW2440, Philips, Eindhoven, The Netherlands) to determine the elemental composition of the samples. 

#### 2.3.4. Fourier Transform Infrared Spectroscopy in Attenuated Total Reflectance Mode (FTIR-ATR)

Thirty-two scans were performed on powder samples from 4000 to 500 cm^−1^ with a resolution of 4 cm^−1^ (Thermo Scientific Nikolet Spectrometer, Unicam Ltd., Ilminster, UK). Sixty-four background scans were initially taken with an empty set up and this scan was then subtracted from all subsequent scans. 

#### 2.3.5. Radiopacity

SrHA powder (0.1 g) was placed in a 96-well plate in triplicate for each sample produced. An X-ray image was taken with 0.25 s exposure time and developed in a dark room alongside an aluminium step wedge in order to express the radiodensity of the materials in mm of aluminium. The films were scanned and analysed using Quantity One software to obtain quantitative values for the radiopacity of the materials. 

### 2.4. In Vitro Biocompatibility of SrHA Pastes and Gels

MG63 human osteosarcoma cell line was cultured in T75 flasks in the following media (*v/v*%): 87% minimum essential media Eagle’s α-modification (α-MEM), 10% foetal calf serum (FCS, Labtech International Ltd., Heathfield, UK), 1% penicillin–streptomycin, 1% L-glutamine and 1% non-essential amino acids. When 70–80% confluent, the flasks were washed twice with 10 mL phosphate buffered saline (PBS) used each time. Trypsin-EDTA (2 mL) was added and the flasks were incubated for 5 min at 37 °C, 5% CO_2_. The detachment of the cells from the flasks was monitored using light microscopy (Eclipse TS100, Nikon, Tokyo, Japan). When the cells were detached, 4 mL FCS was added to neutralise the trypsin. The cell suspension was placed in a tube and centrifuged for 5 min at 1000 rpm. The supernatant was carefully removed and the cell pellet was resuspended in an appropriate amount of media for cell counting using a haemocytometer. The cells were then seeded for experiments or seeded for further culture in T75 flasks. 

The protocols for direct and indirect biocompatibility were based on the ISO 10993-5 biocompatibility testing standard where appropriate [26]. The protocol for indirect biocompatibility was also advised by an internal communication from the IMCOSS project (FP7-SME-315679).

#### 2.4.1. Direct Biocompatibility of SrHA Pastes and Gels

For direct biocompatibility assays, 0.1 mL of paste or gel (0, 2.5, 5, 10, 50, and 100 at.% Sr) was placed into a 24-well plate in triplicate. Direct cellular viability was also tested for ReproBone^®^ *novo* (Ceramisys Ltd., Sheffield, UK) using the same protocol alongside the paste or gel samples. A 1 mL cell suspension containing 50,000 MG63 cells was then added to each well alongside control wells containing no material. Paste/gel wells with no cells present were also prepared to act as material controls. The 24-well plates were then incubated for 48 h at 37 °C, 5% CO_2_. After 48 h, light microscopy (Eclipse TS100, Nikon) was used to image cells growing alongside the paste/gel materials. Cellular viability was then tested using a PrestoBlue^®^ assay. To prevent the disruption and loss of paste or gel from the wells, 100 µL PrestoBlue^®^ reagent (Fisher Scientific, Loughborough, UK) was directly added to the wells, after 100 µL media had been removed. After 1 h, 200 µL of the solution was transferred to a 96-well plate. The fluorescence of the samples was read on a fluorescent plate reader (Infinite M200, Tecan, Reading, UK) with 535 nm excitation and 590 nm emission wavelengths used. After the PrestoBlue^®^ assay the wells were washed twice with 1 mL PBS, and 400 µL of live/dead stain, prepared according to the manufacturer’s instructions, was added to each well and incubated for 15 min at 37 °C, 5% CO_2_. The wells were then washed with PBS before being imaged using a fluorescence microscope (Axiovert 200M, Zeiss, Cambridge, UK). Dead cells could not be imaged for the direct biocompatibility assay due to red background fluorescence from residual PrestoBlue^®^ media solution, which had absorbed into the pastes and gels. The experiment was repeated three times. All PrestoBlue^®^ assays results were expressed as the mean of the three triplicate repeats (*n* = 9). The results were tested for statistical significance using a one-way ANOVA test with Tukey’s multiple comparisons test applied using GraphPad Prism 6 software. 

#### 2.4.2. Indirect Biocompatibility of SrHA Pastes and Gels

Indirect biocompatibility was investigated by incubating 0.5 mL of SrHA paste/gel material (0, 2.5, 5, 10, 50 and 100 at.% Sr) with 5 mL media for 24 h at 37 °C to produce pre-conditioned media. Pre-conditioned media for a commercially available nHA paste (ReproBone^®^ *novo*, Ceramisys Ltd.) was prepared following the same procedure, and 5 mL of media was also incubated without any material present for a non-material control. At the same time, 50,000 MG-63 human osteoblast-like cells in 1 mL media were seeded in a 24-well plate in triplicate for each sample and incubated for 24 h. After 24 h incubation, the pre-conditioned media was centrifuged at 1000 rpm for 5 min. The media on the cells was replaced by 1 mL pre-conditioned media and the cells were incubated again for 24 h. Light microscopy was conducted in the same manner as described above to assess the morphology of the cells cultured in the pre-conditioned medium. A PrestoBlue^®^ assay was carried out by removing the pre-conditioned media and adding 1 mL 10 *v/v*% dilution of the PrestoBlue^®^ in complete media to each well, with the assay, live/dead staining and statistical analysis conducted in the same manner as described above. The experiment was repeated three times, with the statistical analysis performed using a one-way ANOVA as described above for the direct biocompatibility results.

## 3. Results

### 3.1. Materials Characterisation 

#### 3.1.1. X-ray Diffraction 

The addition of Sr caused a peak shift to lower degrees 2θ in the XRD patterns for the apatite phases produced when greater amounts of Sr were used for both the rapid-mixing wet precipitation and sol-gel methods (Figure 1). This peak shift to lower degrees 2θ was also observed for the minor phase of Sr-substituted β-tricalcium phosphate that was present for the apatite produced using the wet precipitation method after high-temperature treatment (Figure 1B). High thermal stability was observed for the samples prepared using the wet precipitation method, i.e., low amounts of decomposition of the hydroxyapatite into β-tricalcium phosphate / β-tristrontium phosphate. However, the Sr-substituted samples prepared using the sol-gel method displayed decomposition, which was in contrast to the Ca nHA prepared using this method (Figure 1D). Therefore, there was no single sintering temperature that could be employed to increase the sharpness of the diffraction peaks of the entire range of SrHA produced using the sol-gel method, without causing excessive thermal decomposition.

#### 3.1.2. Transmission Electron Microscopy 

Particles of 100 at.% SrHA, produced using both methods, had a noticeably larger aspect ratio than any other level of Sr substitution (Figure 2 and Figure 3). The samples were generally homogenous particles, with smaller rounded particles observed for 0 at.% Sr particles compared to elongated particles obtained at 100 at.% Sr. For the wet precipitation method, the aspect ratio increased from 1.7 for 0 at.% Sr to 4.5 for 100 at.% Sr. In detail, the average particle size increased from 50 × 30 nm for 0 at.% Sr to 90 × 20 nm for 100 at.% Sr. The increase in the aspect ratio for the particles produced using the sol-gel method was even more pronounced, increasing from 2.5 for 0 at.% Sr to 6 for 100 at.% Sr. Specifically, for the sol-gel method, the average particle size increased from 50 × 20 nm for 0 at.% Sr to 120 × 20 nm for 100 at.% Sr. For the wet precipitation method, the particles produced with 2.5, 5, 10 and 50 at.% Sr appeared to be composed of smaller particles, which were agglomerated together. This was in contrast to the particles produced at 0 and 100 at.% Sr, which had well-defined edges. 

#### 3.1.3. X-ray Fluorescence

XRF data showed the successful incorporation of Sr into the product, with the product Sr/(Sr + Ca) close to the intended substitution level (Table 1 and Table 2). However, the (Ca + Sr)/P ratios were lower than the stoichiometric value (1.67). For each method, the sample composed of Ca HA had the (Ca + Sr)/P ratio closest to the stoichiometric value (1.63 and 1.59 for the wet precipitation and sol-gel methods, respectively). The samples containing Sr had (Ca + Sr)/P ratios lower than these values. 

#### 3.1.4. Fourier Transform Infrared Spectroscopy in Attenuated Total Reflectance Mode

The FTIR-ATR spectra (Figure 4) for the range of SrHA powders produced by both methods displayed the presence of the following characteristic bands for hydroxyapatite: 3750 cm^−1^ (OH^−^ stretch ν_OH_); 1086 and 1022 cm^−1^ (PO_4_^3−^ ν_3_); 962 cm^−1^ (PO_4_^3−^ ν_1_); 630 cm^−1^ (OH^−^ libration δ_OH_); 600 and 570 cm^−1^ (PO_4_^3−^ ν_4_). Additional band allocation for the unsintered samples were as follows: broad peak centred around 3400 cm^−1^ (absorbed water molecules); 1455 and 1410 cm^−1^ (CO_3_^2−^ ν_3_); 880 cm^−1^ (CO_3_^2−^ ν_2_). The band at around 880 cm^−1^ may also be associated with hydrogen phosphate (P-OH ν_2_), but it is generally accepted that the strong CO_3_^2−^ ν_2_ band obscures this activity [27]. The absorbed water and carbonate groups (Figure 4A) were removed during high-temperature sintering for the wet precipitation samples (Figure 4B). There was a clear shift in the position of the phosphate group bands (1086, 1022 and 962 cm^−1^) to a lower wavenumber when an increased amount of Sr was incorporated for both methods (Figure 5 and Appendix A). For the wet precipitation method, shoulders on the phosphate group bands were observed for the sintered samples (Figure 5B), which corresponded to the presence of β-tricalcium phosphate (β-TCP) or β-tristrontium phosphate (β-TSP) [28]; the approximate positions were 990 cm^−1^ (for 0 at.% Sr), 974 cm^−1^ (for 2.5 and 5 at.% Sr) and 1023 cm^−1^ (for 100 at. % Sr). The bands for the phosphate group activity for the 50 at.% Sr nHA were not as clearly defined as the other Sr contents. This was observed for all the samples analysed. The carbonate bands for the 50 and 100 at.% SrHA samples produced by both methods were also less defined when compared to the other Sr contents (Figure 5C). 

#### 3.1.5. Radiopacity 

The radiopacity of the SrHA powders increased with increasing Sr content (Figure 6A) in a non-linear manner (Figure 6B). The radiopacity values recorded were slightly higher for the powders produced using the sol-gel method (ranging from 3.4 to 7.7 mm of Al) than those found for the powders produced using the wet precipitation method (ranging from 2.0 to 6.6 mm of Al). However, the relationship between Sr at.% and radiopacity was consistent between the samples produced using the two different methods. 

### 3.2. In Vitro Biocompatibility

#### 3.2.1. Direct Biocompatibility

Direct biocompatibility assays were affected by the disaggregation of pastes and gels in culture. Specifically, the 5 at.% SrHA paste and the majority of the SrHA gel samples disaggregated during culture. Excessive disaggregation contributed to a reduced surface area of tissue culture plastic for cells to proliferate. Furthermore, the presence of disaggregated paste and gel in the media may have disrupted the attached cells, as fragments pass over cells whilst the plate was handled. Therefore, the assessment of direct and indirect biocompatibility is appropriate in order to separately consider the biocompatibility of the materials as well as any changes caused by the materials on tissue culture media (i.e., changes in pH or the presence of elutes).

Light microscopy and fluorescence microscopy showed the survival and morphology of the cells in direct contact with the paste (Figure 7 and Figure 8). For the fluorescence microscopy images, some cells close to the paste had a rounded morphology, which may be due to the growth of cells inside the pastes (Figure 8). The cells cultured alongside the SrHA pastes did not have a significantly different viability to the cells cultured alone (Figure 9A). However, the highest viability was observed in the cells cultured with 0 at.% SrHA paste, and the lowest viability was observed with the paste containing 5 at.% SrHA.

The cells cultured with 0 and 100 at.% SrHA gels had significantly reduced viability compared with the cells cultured on tissue culture plastic (Figure 9B). The cells cultured with 50 at.% SrHA gel had the highest viability of the experimental gels tested in direct culture. The SrHA gels were observed to disaggregate considerably during the 48 h incubation period. Microscopy images displayed the growth of cells in direct contact with gel fragments (Figure 7). Similarly to the live stain fluorescence images for the SrHA pastes, cells with a rounded morphology were observed at the edges of the gels, which may have been evidence of cells proliferating within the gels (Figure 8).

The comparison of the direct biocompatibility results for the SrHA pastes and gels (Figure 9A,B) showed no consistent correlation between the amount of Sr in the materials and the viability of cells cultured alongside them, with the highest direct viability results for the pastes obtained with 0 at.% SrHA compared to 50 at.% SrHA for the gels.

#### 3.2.2. Indirect Biocompatibility

For the indirect assays, the 2.5, 5 and 10 at.% SrHA paste showed a significantly lower viability than the cells cultured on tissue culture plastic (Figure 9C). Moreover, the 0 and 100 at.% SrHA pastes showed similar levels of viability as the cells cultured on tissue culture plastic. A variable number of SrHA particles were observed in the pre-conditioned media between the different materials tested, despite all media undergoing centrifugation prior to application. Live/dead fluorescence images showed that the majority of cells were alive, with few dead cells observed (Figure 10).

Pre-conditioned media, produced by incubation with the Sr gels, had no detrimental effect on the viability of the cells (Figure 9D). Also, nHA particles were observed in the gel pre-conditioned media after centrifugation, as observed for the paste pre-conditioned media. Live/dead fluorescence images showed good cell survival, with few dead cells observed (Figure 10).

Viability data for the indirect paste and gel cultures followed a similar pattern, with a reduction in viability observed from 0 at.% to 5 at.% SrHA. The viability then increased for the cells cultured indirectly with the 5 at.% to 100 at.% SrHA pastes and gels. The highest levels of viability were observed for the 100 at.% SrHA paste and gel for the respective ranges of materials.

## 4. Discussion

The substitution of Sr for Ca in the HA crystal lattice had a variety of physicochemical effects, which were detected using a comprehensive range of materials characterisation techniques. Firstly, the increase in unit cell parameters for SrHA was evident in the peak shift to lower degrees 2θ displayed in XRD patterns (Figure 1). This can be attributed to the larger ionic radius of Sr (0.12 nm) when compared to Ca (0.099 nm), and causes a linear unit cell increase over the whole range of Sr substitution (0–100 at.%) [17]. Sr is able to substitute into both Ca sites within the HA crystal lattice, with some reports suggesting a preference for Sr substitution into the M(II) site, particularly for Sr levels higher than 5 at.% [16,17,29]. The M(II) site has been described as ‘larger’ than M(I) to signify the increased preference for the larger Sr ion to occupy this site. This is due to the ability of the M(II) atoms to form staggered equilateral triangles, compared to the constrained columns of the M(I) sites aligned parallel to the c-axis [16].

SrHA produced using the wet precipitation method displayed greater thermal instability with the co-presence of Sr and Ca in the crystal, with the lowest intensity breakdown product peaks observed for 100 at.% Ca HA and 100 at.% SrHA (Figure 1B). This is likely to be due to strain exerted on the crystal lattice, due to the larger ionic radius of Sr when compared to Ca. This contributes to disorder within the crystals, with mixed Sr and Ca HA products generally displaying smaller crystallite sizes [16,17]. This can be observed by the broadening of the XRD peaks, with the unsintered 50 at.% SrHA samples produced using both methods having the broadest peaks (Figure 1A,C). Further evidence for the reduced crystallite size of the Sr and Ca mixed apatite was observed in the FTIR-ATR spectra (Figure 4 and Figure 5), where the phosphate group bands were broader for the 50 at.% SrHA compared with all the other SrHA compositions produced using both methods. This effect has also been correlated with a decreased product crystallite size and crystallinity [30,31].

The noticeable difference between the thermal stability of the SrHA produced by the different methods (Figure 1) could not be attributed to Ca/Sr deficiency, as similar (Ca + Sr)/P ratios (1.50 and 1.52) were obtained for the 100 at.% SrHA produced using the wet precipitation and sol-gel methods, respectively (Table 1 and Table 2). It has been reported that the presence of B-type carbonate substitution stabilises the apatite structure, due to the presence of water in two types of vacancies [32]. The vacancies are present due to the negative charge deficiency when the CO_3_^2−^ ion substitutes for the PO_4_^3−^ ion [33]. Therefore, B-type carbonate substitution may have contributed to the increased thermal stability for the 100 at.% SrHA produced using the wet precipitation method. Although the B-type carbonate substitution bands were less defined for 50 and 100 at.% SrHA produced using both methods, it can be seen that the sol-gel samples showed a flatter profile in this region (Figure 5D) than the wet precipitation materials (Figure 5C). Additionally, from the TEM micrographs, it can be observed that the 100 at.% SrHA produced using the sol-gel method (Figure 3) had much straighter edges and sharper vertices than the 100 at.% SrHA produced using the wet precipitation method (Figure 2). This could suggest a higher surface energy of the sol-gel 100 at.% SrHA, which in turn could contribute to its lower thermal stability.

The TEM images clearly displayed the noticeable effect Sr had on the crystal growth of the SrHA particles. Although much more pronounced for the sol-gel method (Figure 3), it was still clear for the wet precipitation method that the 100 at.% SrHA was composed of particles with a higher aspect ratio than the 100 at.% Ca nHA (Figure 2). This change in particle aspect ratio suggested that the presence of Sr caused a change in the precipitation rate. Specifically, growth along the c-axis may be promoted due to the energy required for crystal growth being lower than the energy required for new crystal nucleation [34]. The mixed Ca–Sr nHA particles were generally composed of smaller sub particles with ill-defined edges, particularly for the wet precipitation method. Similar effects were reported elsewhere and corresponded with a lower degree of crystallinity [16].

The XRF results demonstrated that the level of incorporation of Sr was generally very close to the attempted substitution level, albeit in a Ca/Sr deficient product (Table 1 and Table 2). This showed the readiness of Sr to substitute into the nHA crystal lattice, aided by the isoelectric nature of the substitution of divalent Ca ions for divalent Sr ions.

The presence of Sr in the nHA had a clear effect on the phosphate group bands, as shown by the peak shift to a lower wavenumber, observed in the FTIR-ATR spectra (Figure 5). This has been attributed to the larger Sr atom altering the lattice vibrational energies [16]. The FTIR-ATR spectra showed modest levels of carbonate substitution for the SrHA produced using both methods (Figure 5C,D). The bands at around 1450 and 1420 cm^−1^ can be attributed to the B-type substitution of carbonate groups for phosphate groups. A possible explanation for the carbonate substitution may be that the alkalinity of the precipitation solutions allowed for dissolution of atmospheric carbon dioxide. The carbonate groups observed for the wet precipitation method products were removed during the sintering treatment at 1000 °C (Figure 4B).

The clear increase in radiopacity with increasing Sr content was expected, due to the higher atomic mass of Sr compared to Ca (Figure 6). This allows Sr to absorb more X-ray energy [35]. The density of the different powders may have contributed to the difference seen between the powders produced by the different methods, as for the same Sr content, a higher radiopacity was observed for the sol-gel powders. The difference in density may be linked to the amount of absorbed water present in the powders.

The in vitro biocompatibility results highlighted the sensitivity of cultured cells to different forms of SrHA-based materials, alongside the effects of different exposure routes. Indeed, any differences in cellular viability observed in these experiments (Figure 9) may have been due to a combination of effects arising from changes in HA chemistry, particle morphology, culture configuration, or the water content of the SrHA pastes and gels. Furthermore, disaggregation of the materials in tissue culture media is likely to have contributed to some of the effects observed. In detail, SrHA gels consistently disaggregated when submerged in tissue culture media, which may have allowed for more standardised pre-conditioned media preparation for the indirect assays. In contrast, the 5 at.% SrHA paste displayed the greatest disaggregation, thus increasing the surface area of exposure during the preparation of the preconditioned media, and reducing the surface area in the well plate where cells could proliferate in an unconstrained manner for direct cultures. These effects are likely to have contributed to the decreased viability of the cells cultured directly and indirectly with this material. Therefore, the indirect biocompatibility results (Figure 9C,D) may provide a more relevant insight than the direct biocompatibility results (Figure 9A,B), due to the higher potential of the direct biocompatibility results being dependent on these culture condition artefacts.

SrHA is known to have increased solubility compared to Ca nHA, with the increase in solubility proportional to Sr content [13]. Therefore, it could be suggested that increased solubility caused a higher concentration of dissolved ions, which may have affected cellular viability through osmotic effects or other mechanisms. The presence of the larger Sr ion has been proposed to contribute to the increased solubility of SrHA samples, due to a destabilising effect on the crystal structure [13].

The difference between biocompatibility investigations observed for the pastes and gels may have been affected by the different water contents of the two materials, since the gel was composed of 90 wt.% water and the paste had an 80 wt.% water content. Therefore, the SrHA paste cultures would have been subjected to a higher dose of nHA particles and presumably a higher dose of any ions released from the materials. Conversely, the gel materials had a greater potential to release more water into the media, which may have affected the cellular response through osmotic effects.

For the various materials and culture configurations tested, there was no consistent correlation between cellular viability and Sr content, suggesting that the increased amount of Sr or changes in particle morphology were not detrimental to the viability of the cells (Figure 9). With the exception of the cells cultured directly with the SrHA gels, the cellular viability did not fall below 77%. This demonstrated the low cytotoxicity of the materials tested, with a reduction in cellular viability to 70% or lower considered as cytotoxic in ISO 10993-5:2009 [26]. As mentioned above, the disaggregation of the gels in vitro was likely to contribute to a reduction in viability, particularly for the direct gel assays.

Existing published reports on the biocompatibility of SrHA have concluded a variety of optimal Sr concentrations. It must be noted that different Sr content ranges are often considered, which makes it difficult to compare between studies. Alongside this, investigations of different forms of material and different particle morphologies can further complicate the results, due to their distinct effects on cellular behaviour. Several experiments with SrHA tested on MG63 cells have reported that the presence of Sr has a positive effect on cellular viability [8,36] and early expression of alkaline phosphatase [37]. In contrast, other investigations have found that SrHA has no significant effect on cellular viability for MSCs [9] or for L929 fibroblasts [21]. These results highlight the limitations of in vitro biocompatibility testing, as different results may be obtained using different cells types or culture configurations. For instance, Crawford et al. have reported that nHA particles have a cytotoxic effect for a range of cell types, with two-dimensional cultures showing increased sensitivity compared to three-dimensional systems [38]. Therefore, it is necessary to test promising compositions in vivo in order to fully realise the bone regeneration potential of these materials in a representative defect model.

This work has demonstrated the successful production of SrHA pastes and gels using relatively simple, rapid-mixing, low-cost methodologies. The current evidence suggests that these materials are likely to have a beneficial biological response. In particular, the detailed characterisation has provided evidence of SrHA being produced with bioinspired features, including nanoscale dimensions, carbonate substitution, and Ca deficiency, all of which may contribute to increased biological activity. Therefore, this study is of great value in informing the development of the next generation of injectable nHA materials. Combined with the findings of previous studies, this work demonstrates the potential to make further inorganic modifications to generate multi-action biomaterials; for example the combination of antibacterial properties with enhanced osteoconductivity to produce a biomaterial to simultaneously combat bone infections and promote optimal bone tissue regeneration.

## 5. Conclusions

Here, we present the first detailed study of the successful synthesis, characterisation and side-by-side comparison of SrHA pastes and gels containing 0 to 100 at.% Sr, fabricated using two different methodological approaches, rapid-mix wet precipitation and sol-gel. Both methods produced a consistent and predictable incorporation of Sr into the final product. In both cases, SrHA produced with 50 at.% Sr showed characteristics associated with having the lowest crystallite size in the range of the materials produced. When SrHA was fully substituted (i.e., 100 at.%), the particle aspect ratio was increased, most likely due to altered precipitation kinetics. SrHA had greater radiopacity that was proportional to its Sr content, which is a property with specific clinical benefits related to the detection of material during deployment and monitoring post-surgery. In vitro biocompatibility tests suggested that SrHA pastes and gels supported the growth of osteoblastic-like cells with direct and indirect material contact, with comparable biocompatibility to a non-substituted commercial bone graft substitute reference material. While the products from the two methods were consistent, it was concluded that the rapid-mix wet precipitation was better suited to translation to industry on account of its relative simplicity and proven versatility. The Sr substitution accuracy using this method was also higher, with measured Sr levels within +/− 0.5 at.% Sr across the whole range of substitutions attempted. The innovative injectable materials reported here are potentially superior to current bone graft biomaterials on account of assisting minimally invasive delivery and successful strontium incorporation, which has the potential to impart greater radiopacity and improved stimulation of bone tissue regeneration.

## Figures and Tables

**Figure 1 nanomaterials-11-01611-f001:**
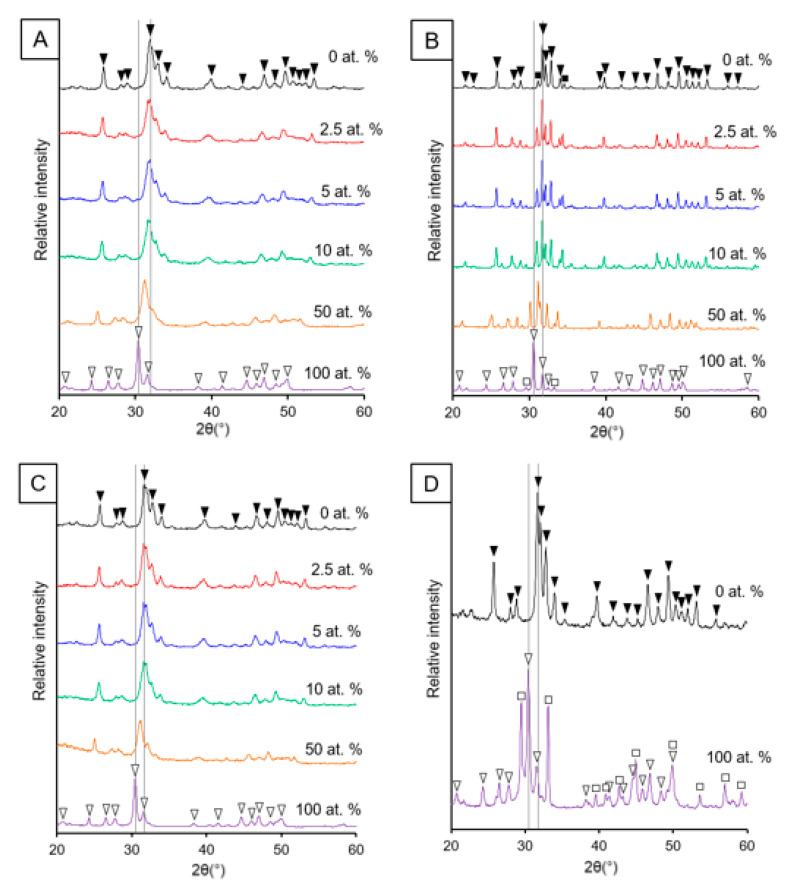
XRD patterns of SrHA powders (0, 2.5, 5, 10, 50 and 100 at.% Sr) produced using wet precipitation (**A**,**B**) and sol-gel (**C**,**D**) methods. Unsintered powders (**A**,**C**) and powders sintered at 1000 °C (**B**) and 700 °C (**D**). Peak labels: 
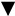
 hydroxyapatite; 
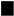
 β-tricalcium phosphate; 
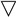
 strontium hydroxyapatite; 
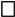
 β-tristrontium phosphate.

**Figure 2 nanomaterials-11-01611-f002:**
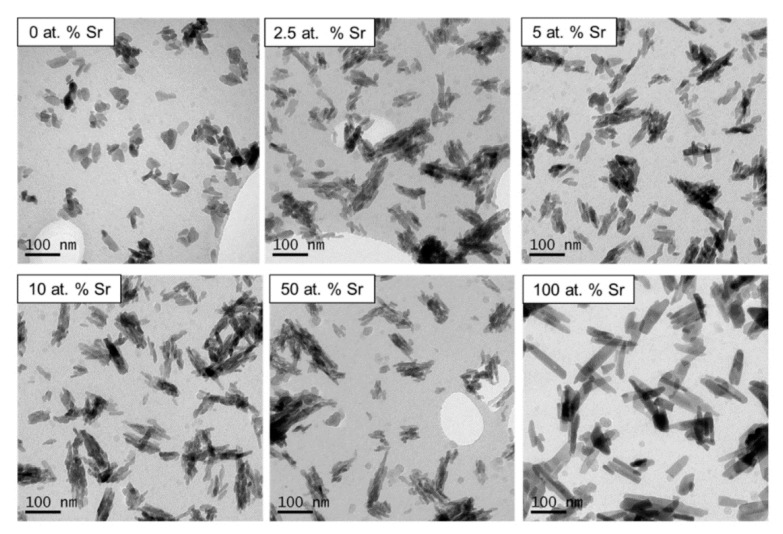
TEM micrographs of unsintered SrHA powders (0, 2.5, 5, 10, 50 and 100 at.% Sr) produced using the wet precipitation method.

**Figure 3 nanomaterials-11-01611-f003:**
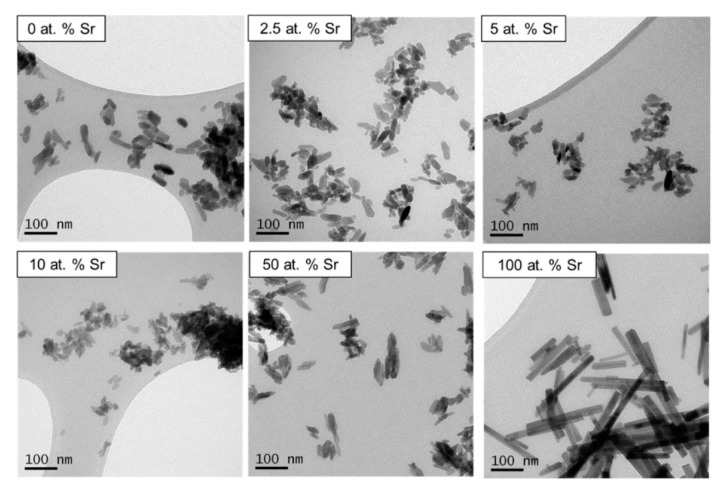
TEM micrographs of unsintered SrHA powders (0, 2.5, 5, 10, 50 and 100 at.% Sr) produced using the sol-gel method.

**Figure 4 nanomaterials-11-01611-f004:**
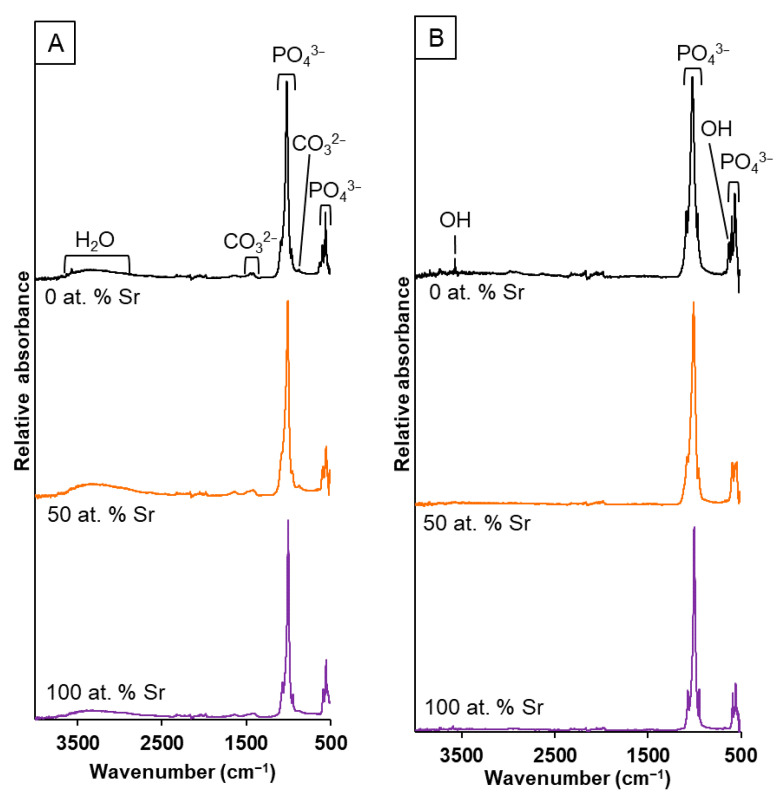
FTIR-ATR spectra of SrHA powders (0, 50 and 100 at.% Sr) produced using the wet precipitation method: unsintered (**A**) and sintered at 1000 °C (**B**).

**Figure 5 nanomaterials-11-01611-f005:**
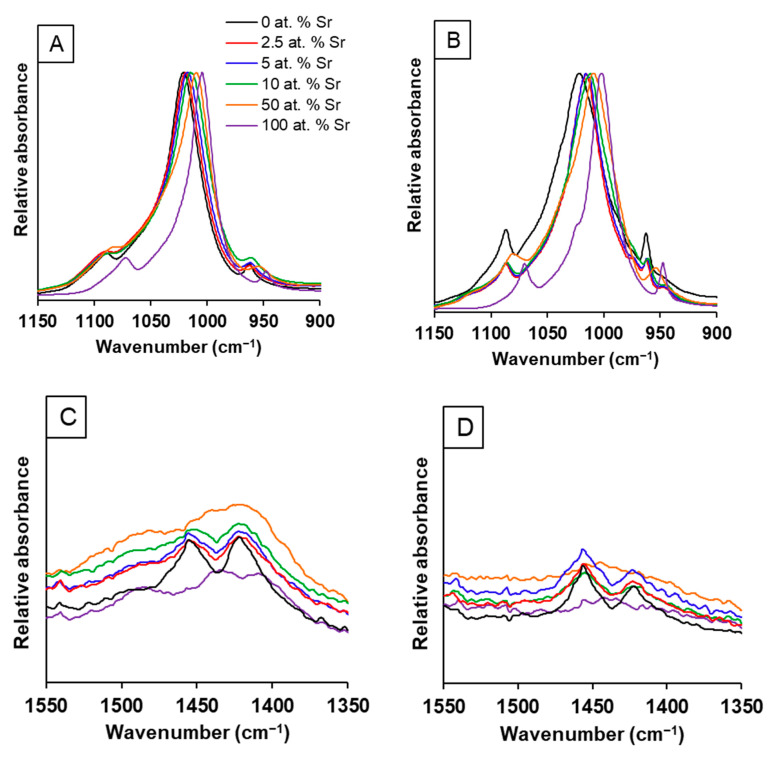
FTIR-ATR spectra of SrHA powders (0, 2.5, 5, 10, 50 and 100 at.% Sr) produced using the wet precipitation method. Phosphate peaks of unsintered (**A**) and sintered (**B**) powders. Carbonate peaks of unsintered powders from wet precipitation (**C**) and sol-gel (**D**) methods. Legend for all graphs as shown in A.

**Figure 6 nanomaterials-11-01611-f006:**
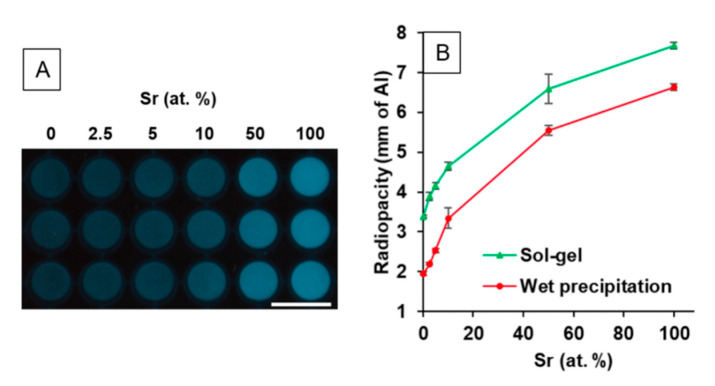
(**A**) Radiograph of SrHA powders (0, 2.5, 5, 10, 50 and 100 at.% Sr) produced using the wet precipitation method, *n* = 3. Scale bar = 1 cm. (**B**) Radiopacity of SrHA powders (0, 2.5, 5, 10, 50 and 100 at.% Sr) produced using sol-gel and wet precipitation methods, *n* = 3.

**Figure 7 nanomaterials-11-01611-f007:**
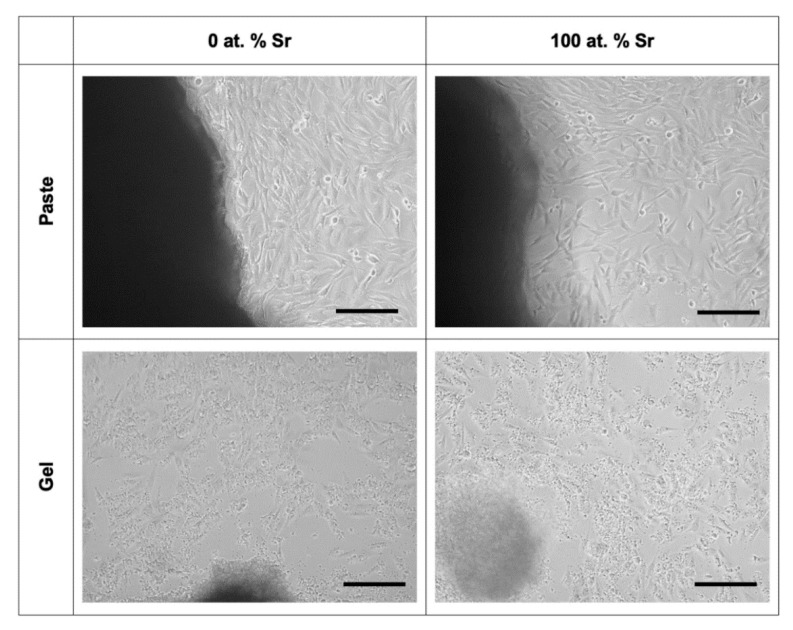
Light microscopy images for MG63 cells cultured directly with SrHA paste and gel (0 and 100 at.% Sr). Scale bar = 200 µm.

**Figure 8 nanomaterials-11-01611-f008:**
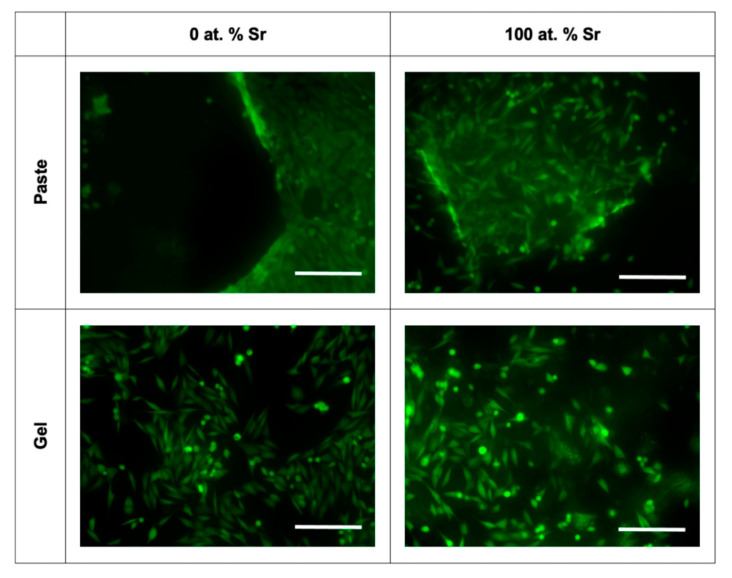
Live fluorescence imaging for MG63 cells cultured directly with SrHA paste and gel (0 and 100 at.% Sr). Scale bar = 200 µm.

**Figure 9 nanomaterials-11-01611-f009:**
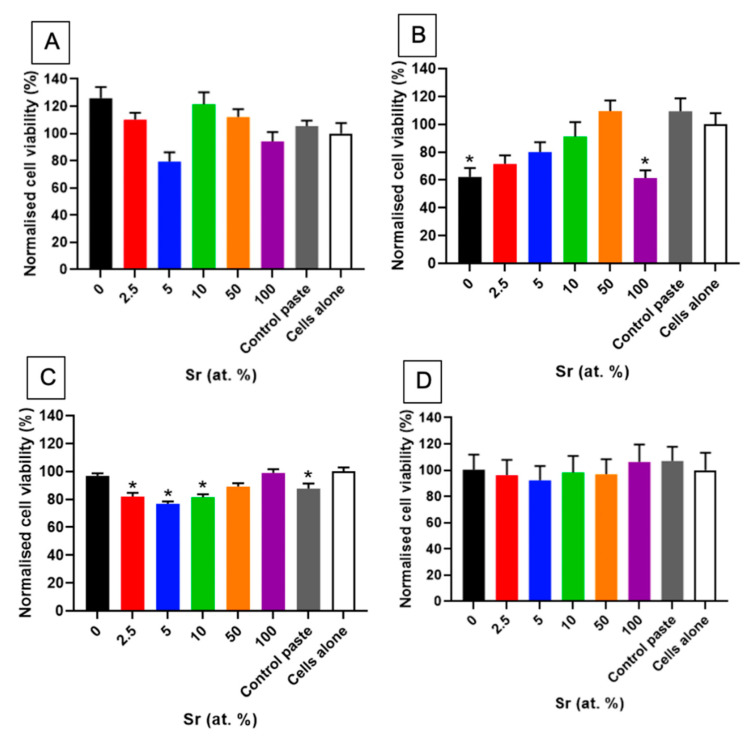
PrestoBlue^®^ viability assay results for MG63 cells cultured directly for 48 h with SrHA (0, 2.5, 5, 10, 50 and 100 at.% Sr) pastes (**A**) or gels (**B**) or indirectly for 24 h with SrHA pastes (**C**) or gels (**D**). ReproBone^®^ *novo* used as control paste alongside cells with no material present (*n* = 9). Error bars ± standard error of the mean. Significance level compared to cells alone control * *p* < 0.05.

**Figure 10 nanomaterials-11-01611-f010:**
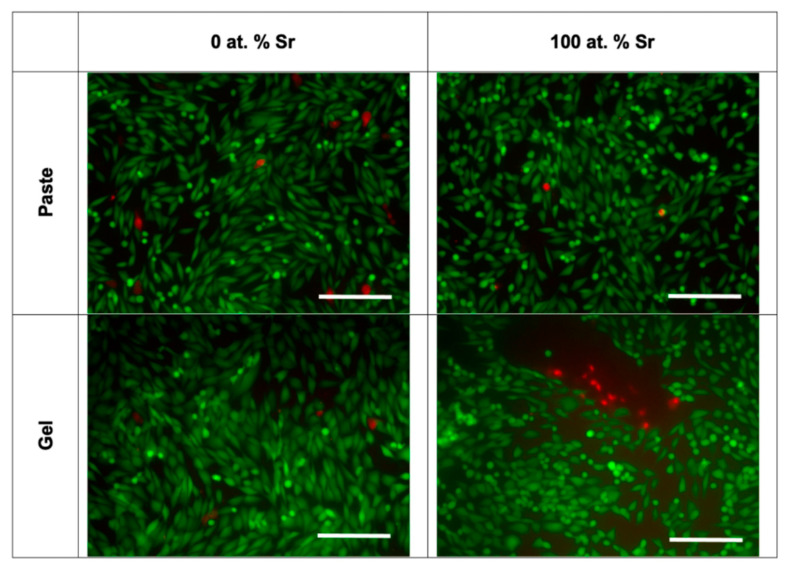
Live/dead fluorescence imaging for MG63 cells cultured indirectly with SrHA paste and gel (0 and 100 at.% Sr). Scale bar = 200 µm.

**Table 1 nanomaterials-11-01611-t001:** Reagent amounts used to produce SrHA (0, 2.5, 5, 10, 50 and 100 at.% Sr) using rapid-mixing wet precipitation method. Strontium incorporation and (Ca + Sr)/P molar ratio achieved, characterised using X-ray fluorescence.

Amount of Sr (at.%)	Material Preparation				Characterisation Results	
	Calcium Hydroxide Amount		Strontium Hydroxide Octahydrate Amount			
	g	mmol	G	mmol	Sr / (Sr + Ca) at.%	(Ca + Sr)/P Molar Ratio
0	3.70	50	0	0	0.02	1.63
2.5	3.61	48.75	0.33	1.25	2.45	1.57
5	3.52	47.5	0.66	2.5	4.83	1.57
10	3.33	45	1.33	5	9.55	1.56
50	1.85	25	6.64	25	49.67	1.58
100	0	0	13.29	50	99.79	1.50

**Table 2 nanomaterials-11-01611-t002:** Reagent amounts used to produce SrHA (0, 2.5, 5, 10, 50 and 100 at.% Sr) using rapid-mixing sol-gel method. Strontium incorporation and (Ca + Sr)/P molar ratio achieved, characterised using X-ray fluorescence.

Amount of Sr (at.%)	Material Preparation				Characterisation Results	
	Calcium Nitrate Tetrahydrate Amount		Strontium Nitrate Amount			
	g	mmol	g	mmol	Sr/(Sr + Ca) at%	(Ca + Sr)/P Molar Ratio
0	11.81	50	0	0	0.01	1.59
2.5	11.51	48.75	0.26	1.25	2.32	1.56
5	11.22	47.5	0.53	2.5	4.81	1.50
10	10.63	45	1.06	5	9.01	1.54
50	5.90	25	5.29	25	45.11	1.48
100	0	0	10.58	50	99.91	1.52

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
