# Peer review of "Nanoscale Strontium-Substituted Hydroxyapatite Pastes and Gels for Bone Tissue Regeneration"

_nanomaterials, 2021, doi:10.3390/nano11061611_

Round 1

Reviewer 1 Report

This is a nice publication carefully put together, well-written and well explained, with lots of work and experiments having been done and lots of useful data. The obtained results are supported by experimental evidences. The study could be of interest for the readers of the journal Nanomaterials. However, the authors need to explain the fundamental difference between their work and all previous studies (relevant, recent studies are not presented in the Introduction section).

Abstract is relatively broad and did not point out the main findings and the novelty of the work. In this regard, the abstract shall specify the summary of the article with numbers rather than general discussion. The conclusion section should have the main results in quantitative statements as well.

Materials and methods. Please standardize the details regarding all used equipment (such as name, model, manufacturer, measurement conditions, etc.) for measurements. No details regarding the used chemicals are provided (such as provenience, purity).

Results and discussion. Please use the chemical symbol of elements instead of the name (example: Sr – strontium). The Results and discussion section should be better presented in order to highlight the most significant (and unexpected results), find correlations, patterns and relationships among the data, speculations, limitations of work and deductive arguments. Also, the authors should introduce more studies related to their work and to correlate the obtained results according to previous/ similar / many studies.

The results obtained are interesting and promising. The manuscript can be accepted for publication in Nanomaterials journal after MAJOR corrections.

Author Response

Response to reviewer 1

This is a nice publication carefully put together, well-written and well explained, with lots of work and experiments having been done and lots of useful data. The obtained results are supported by experimental evidences. The study could be of interest for the readers of the journal Nanomaterials. However, the authors need to explain the fundamental difference between their work and all previous studies (relevant, recent studies are not presented in the Introduction section).

We would like the thank the reviewer for their comments and we have added and discussed the following additional studies to address this point: 

Dai, J.; Fu, Y.; Chen, D.; Sun, Z. A novel and injectable strontium-containing hydroxyapatite bone cement for bone substitution: A systematic evaluation. Materials Science and Engineering: C 2021, 124, 112052, doi:10.1016/j.msec.2021.112052.

Yuan, B.; Raucci, M.G.; Fan, Y.; Zhu, X.; Yang, X.; Zhang, X.; Santin, M.; Ambrosio, L. Injectable strontium-doped hydroxyapatite integrated with phosphoserine-tethered poly(epsilon-lysine) dendrons for osteoporotic bone defect repair. Journal of Materials Chemistry B 2018, 6, 7974-7984, doi:10.1039/C8TB02526F.

Abstract is relatively broad and did not point out the main findings and the novelty of the work. In this regard, the abstract shall specify the summary of the article with numbers rather than general discussion. The conclusion section should have the main results in quantitative statements as well.

The abstract states the following with respect to the main findings and novelty of the work:

‘The full range of nanoscale SrHA materials were successfully prepared using both methods, with measured substitution very close to the calculated amounts. As anticipated, the SrHA samples showed increased radiopacity, a beneficial property to aid in vivo or clinical monitoring of the material in situ over time. For indirect methods, the greatest cell viabilities were observed for the 100 % substituted SrHA paste and gel, while direct viability results were most likely influenced by material disaggregation in the tissue culture media.’

‘…..there are no detailed reports describing the preparation of a systematic substitution up to 100% at the nanoscale.’

We have added quantitative statements regarding the Sr substitution accuracy achieved to the conclusion due to the limited word count available in the abstract.

Materials and methods. Please standardize the details regarding all used equipment (such as name, model, manufacturer, measurement conditions, etc.) for measurements. No details regarding the used chemicals are provided (such as provenience, purity).

The equipment names and manufacturers have been standardised as requested. The following text has also been added in section 2: ‘All reagents were obtained from Sigma Aldrich (UK) unless otherwise stated.’ The purity of the reagents used for the preparation of the materials has also been added in the text in section 2.

Results and discussion. Please use the chemical symbol of elements instead of the name (example: Sr – strontium).

These have been replaced as suggested.

The Results and discussion section should be better presented in order to highlight the most significant (and unexpected results), find correlations, patterns and relationships among the data, speculations, limitations of work and deductive arguments. Also, the authors should introduce more studies related to their work and to correlate the obtained results according to previous/ similar / many studies.

Reviewer 2 Report

This is a well-presented manuscript describing the synthesis of nanoscale strontium-substituted hydroxyapatite paste by a rapid precipitation method and nanoscale strontium-substituted gel by a sol-gel technique. The authors varied the atomic % strontium between 0 and 100% and characterized the materials produced by a range of different experimental techniques, i.e., X-ray diffraction, X-ray fluorescence, FTIR-spectroscopy, radiopacity, biocompatibility. The manuscript is important for the field because of the potential of the materials synthesized to replace bone in bone surgery.

I have a few comments for consideration by the authors and some minor corrections.

  1. The peak shift of SrHA observed via X-ray diffraction only appears to occur at %at values above 10 (see Fig. 1). Is this worth mentioning?
  2. p. 3, line 108. Insert "of" between "amount" and "strontium".
  3. p. 4, line 132. Omit "were" between "and" and "loaded".
  4. p. 6, lines 259-261. The authors state that SrHA prepared by the sol-gel method displayed decomposition. How is this evident in Figs. 1C and 1D?
  5. p. 6, line 261. The authors state that CanHA prepared by the sol-gel method doesn't undergo thermal decomposition and refer to Fig. 1D. However, the caption of Fig. 1D states that the data shown refers to SrHA, not CaHA. Therefore, there is an inconsistency.
  6. p. 7, line 272. Insert "to" after "contrast".
  7. p. 10, line 298. Replace "closets" with "closest".
  8. The Figures aren't numbered in the order in which they appear in the text, e.g. Fig. 9 is first mentioned on page 13, line 371, whereas Fig. 7 is first mentioned at line 377.
  9. p. 8, line 447, Insert a comma between "site" and "particularly".
  10. p. 8, line 466, Insert a comma between "deficiency" and "as".
  11. p. 8, line 468, Insert a comma in front of "respectively".
  12. p. 8, line 488, Insert "the" in front of "energy".
  13. p. 9, line 533, Insert a "comma" after "Therefore".
  14. p. 9, line 534, Change "effected" to "affected"  

Author Response

Response to reviewer 2

This is a well-presented manuscript describing the synthesis of nanoscale strontium-substituted hydroxyapatite paste by a rapid precipitation method and nanoscale strontium-substituted gel by a sol-gel technique. The authors varied the atomic % strontium between 0 and 100% and characterized the materials produced by a range of different experimental techniques, i.e., X-ray diffraction, X-ray fluorescence, FTIR-spectroscopy, radiopacity, biocompatibility. The manuscript is important for the field because of the potential of the materials synthesized to replace bone in bone surgery.

I have a few comments for consideration by the authors and some minor corrections.

  1. The peak shift of SrHA observed via X-ray diffraction only appears to occur at %at values above 10 (see Fig. 1). Is this worth mentioning?

Although slight, there is peak shift for the strontium substitutions below 10 at. %, as would be expected with this analysis for these materials.

  1. 3, line 108. Insert "of" between "amount" and "strontium".

Inserted

  1. 4, line 132. Omit "were" between "and" and "loaded".

Omitted

  1. 6, lines 259-261. The authors state that SrHA prepared by the sol-gel method displayed decomposition. How is this evident in Figs. 1C and 1D?

Figure 1D displays the significant presence of β-tristrontium phosphate for the 100 at. % SrHA as the result of thermal decomposition. The following text has been added in the results to explain this:

High thermal stability was observed for the samples prepared using the wet precipitation method i.e. low amounts of decomposition of the hydroxyapatite into β-tricalcium phosphate / β-tristrontium phosphate.

  1. 6, line 261. The authors state that CanHA prepared by the sol-gel method doesn't undergo thermal decomposition and refer to Fig. 1D. However, the caption of Fig. 1D states that the data shown refers to SrHA, not CaHA. Therefore, there is an inconsistency.

Figure 1 D is labelled to show calcium hydroxyapatite (0 at. % Sr, top pattern) alongside 100 at. % Sr (bottom pattern). In contrast to the 100 at. % Sr sample, the calcium hydroxyapatite shows no thermal decomposition (no presence of β-TCP) at 700 °C.

  1. 7, line 272. Insert "to" after "contrast".

Inserted

  1. 10, line 298. Replace "closets" with "closest".

Corrected

  1. The Figures aren't numbered in the order in which they appear in the text, e.g. Fig. 9 is first mentioned on page 13, line 371, whereas Fig. 7 is first mentioned at line 377.

The paragraph has been rearranged to describe the results in the order that they are presented.

  1. 8, line 447, Insert a comma between "site" and "particularly".

Inserted

  1. 8, line 466, Insert a comma between "deficiency" and "as".

Inserted

  1. 8, line 468, Insert a comma in front of "respectively".

The authors politely disagree and no change made.

  1. 8, line 488, Insert "the" in front of "energy".

Inserted

  1. 9, line 533, Insert a "comma" after "Therefore".

Inserted

  1. 9, line 534, Change "effected" to "affected"  

Corrected

Reviewer 3 Report

Dear Authors, in your interesting manuscript, the following points should be added/changed to further improve it:

  1. Introduction: I have a comment on the following sentence “Nanoscale hydroxyapatite (nHA) can be considered bioinspired due to its similarity to the mineral found naturally in bone and tooth enamel [2, 3] [42-43]”. The article (doi:10.3762/bjnano.7.153) presents the crystal structure of bone and enamel (XRD results). The authors estimated the average size of hydroxyapatite crystallites in natural bone and enamel. It is worth mentioning in the knowledge review, that there are already methods for the synthesis of hydroxyapatite, which allows to receive a crystal structure practically identical to natural hydroxyapatite.
  2. Introduction: I suggest Authors mention the results of work “Effect of strontium-substituted biphasic calcium phosphate on inflammatory mediators production by human monocytes” (doi:0.1016/j.actbio.2012.04.045)
  3. Materials and Methods: There is no information about the reagents (purity, manufacturer).
  4. Materials and Methods: It is also good practice to provide the chemical formulas of the reagents.
  5. Materials and Methods: Information about the furnace is not provided (model, manufacturer) [125-126].
  6. Materials and Methods - Strontium-substituted nanoscale hydroxyapatite gel preparation using rapid mixing sol-gel method: There are citation errors throughout the manuscript “Error! Reference source not found”.
  7. Materials and Methods - Strontium-substituted nanoscale hydroxyapatite gel preparation using rapid mixing sol-gel method: Please define the word "overnight" [151]. Exactly how many hours it was?
  8. Materials and Methods - Strontium-substituted nanoscale hydroxyapatite gel preparation using rapid mixing sol-gel method: Please explain to me why the authors once sintered the samples at 1000°C [125-126] and the second time at 700°C [153-154]. The authors wrote "Due to excessive thermal decomposition at 1000°C ... [152-153], so please explain to me why in one case they chose a sintering temperature of 1000°C.
  9. Materials and Methods - X-ray fluorescence (XRF): What was the statistic of the method?
  10. Results - X-ray diffraction: Please determine the average crystallite size from the XRD results.
  11. Results - X-ray diffraction: Please explain to me what the expression "High thermal stability [258]" means.
  12. Results: Transmission electron microscopy: Please determine the average particle size from TEM results.
  13. Results: Transmission electron microscopy: Please specify the shape of the particles. Whether the obtained samples were homogeneous?

Author Response

Response to reviewer 3

Dear Authors, in your interesting manuscript, the following points should be added/changed to further improve it:

  1. Introduction: I have a comment on the following sentence “Nanoscale hydroxyapatite (nHA) can be considered bioinspired due to its similarity to the mineral found naturally in bone and tooth enamel [2, 3] [42-43]”. The article (doi:10.3762/bjnano.7.153) presents the crystal structure of bone and enamel (XRD results). The authors estimated the average size of hydroxyapatite crystallites in natural bone and enamel. It is worth mentioning in the knowledge review, that there are already methods for the synthesis of hydroxyapatite, which allows to receive a crystal structure practically identical to natural hydroxyapatite.

This reference has been included in the literature review section as suggested:

‘A variety of methods is available to produce biomimetic nanoscale hydroxyapatite including hydrothermal and wet precipitation [22-24].’

  1. Introduction: I suggest Authors mention the results of work “Effect of strontium-substituted biphasic calcium phosphate on inflammatory mediators production by human monocytes” (doi:0.1016/j.actbio.2012.04.045)

This reference has been discussed in the introduction as suggested:

‘Furthermore it has been shown that the presence of strontium-substituted biphasic calcium phosphate decreased pro-inflamatory cytokine production, suggesting that Sr may assist in controlling inflammatory processes.’ Referenced in paper.

  1. Materials and Methods: There is no information about the reagents (purity, manufacturer).

These details have been added.

  1. Materials and Methods: It is also good practice to provide the chemical formulas of the reagents.

This has also been added.

  1. Materials and Methods: Information about the furnace is not provided (model, manufacturer) [125-126].

This has been included.

  1. Materials and Methods - Strontium-substituted nanoscale hydroxyapatite gel preparation using rapid mixing sol-gel method: There are citation errors throughout the manuscript “Error! Reference source not found”.

These are not present in the word document version of the manuscript. The reference has been reinserted to rectify this problem

  1. Materials and Methods - Strontium-substituted nanoscale hydroxyapatite gel preparation using rapid mixing sol-gel method: Please define the word "overnight" [151]. Exactly how many hours it was?

Approximately 20 hours has been added to define overnight.

  1. Materials and Methods - Strontium-substituted nanoscale hydroxyapatite gel preparation using rapid mixing sol-gel method: Please explain to me why the authors once sintered the samples at 1000°C [125-126] and the second time at 700°C [153-154]. The authors wrote "Due to excessive thermal decomposition at 1000°C ... [152-153], so please explain to me why in one case they chose a sintering temperature of 1000°C.

The high temperature treatment was performed purely to investigate the thermal stability. A sintering temperature of 1000 °C was required to sufficiently increase the sharpness of the diffraction peaks for the product produced using the wet precipitation method, and thus shows the high thermal stability of the products of this method. For the products produced using the sol-gel method, as explained in the result section ‘there was no single sintering temperature that could be employed to increase the sharpness of the diffraction peaks of the entire range of SrHA without causing excessive thermal decomposition.’ The subsequent functional testing on the pastes and gels were all carried out using undried products without a high temperature heat treatment applied.

  1. Materials and Methods - X-ray fluorescence (XRF): What was the statistic of the method?

One sample was analysed per product, therefore there was no measurement error to report.

  1. Results - X-ray diffraction: Please determine the average crystallite size from the XRD results.

  1. Results - X-ray diffraction: Please explain to me what the expression "High thermal stability [258]" means.

The following text has been added in the results to explain this:

‘High thermal stability was observed for the samples prepared using the wet precipitation method i.e. low amounts of decomposition of the hydroxyapatite into β-tricalcium phosphate / β-tristrontium phosphate.’

  1. Results: Transmission electron microscopy: Please determine the average particle size from TEM results.

Average particle sizes have been added in section 3.1.2:

‘In detail, the average particle size increased from 50 x 30 nm for 0 at. % Sr to 90 x 20 nm for 100 at. % Sr.’

‘Specifically, for the sol-gel method the average particle size increased from 50 x 20 nm for 0 at. % Sr to 120 x 20 nm for 100 at. % Sr.’

  1. Results: Transmission electron microscopy: Please specify the shape of the particles. Whether the obtained samples were homogeneous?

These details have been added to section 3.1.2: ‘The samples were generally homogenous particles with smaller rounded particles observed for 0 at. % Sr particles compared to elongated particles obtained at 100 at. % Sr.’

It was also described:

‘For the wet precipitation method the particles produced with 2.5, 5, 10 and 50 at. % Sr appeared to be composed of smaller particles which were agglomerated together. This was in contrast to the particles produced at 0 and 100 at. % Sr which had well defined edges.’

Round 2

Reviewer 1 Report

The manuscript cand be accepted for publication in the present form.

Reviewer 3 Report

The authors addressed most of the comments and questions I formulated during the first round of revision. The revised version of the manuscript is more complete and readable. In this form the manuscript is suitable for publication to me.